# Electrochemical Detection of Dopamine at Fe_3_O_4_/SPEEK Modified Electrode

**DOI:** 10.3390/molecules26175357

**Published:** 2021-09-03

**Authors:** Mogomotsi N. Ranku, Gloria E. Uwaya, Omolola E. Fayemi

**Affiliations:** 1Department of Chemistry, Faculty of Natural and Agricultural Sciences, North-West University (Mafikeng Campus), Private Bag X2046, Mmabatho 2735, South Africa; mogomotsiranku@gmail.com (M.N.R.); okonyeglo@yahoo.com (G.E.U.); 2Material Science Innovation and Modelling (MaSIM) Research Focus Area, Faculty of Natural and Agricultural Sciences, North-West University (Mafikeng Campus), Private Bag X2046, Mmabatho 2735, South Africa

**Keywords:** electrochemical, dopamine, sulphonated polyether ether ketone, Fe_3_O_4_, cyclic and square wave voltammetry

## Abstract

Reported here is the design of an electrochemical sensor for dopamine (DA) based on a screen print carbon electrode modified with a sulphonated polyether ether ketone-iron (III) oxide composite (SPCE-Fe_3_O_4_/SPEEK). *L. serica* leaf extract was used in the synthesis of iron (III) oxide nanoparticles (Fe_3_O_4_NPs). Successful synthesis of Fe_3_O_4_NP was confirmed through characterization using Fourier transform infrared (FTIR), ultraviolet–visible light (UV–VIS), X-ray diffractometer (XRD), and scanning electron microscopy (SEM). Cyclic voltammetry (CV) was used to investigate the electrochemical behaviour of Fe_3_O_4_/SPEEK in 0.1 M of phosphate buffer solution (PBS) containing 5 mM of potassium ferricyanide (III) solution (K_3_[Fe(CN)_6_]). An increase in peak current was observed at the nanocomposite modified electrode SPCE-Fe_3_O_4_/SPEEK) but not SPCE and SPCE-Fe_3_O_4_, which could be ascribed to the presence of SPEEK. CV and square wave voltammetry (SWV) were employed in the electroxidation of dopamine (0.1 mM DA). The detection limit (LoD) of 7.1 μM and 0.005 μA/μM sensitivity was obtained for DA at the SPCE-Fe_3_O_4_/SPEEK electrode with concentrations ranging from 5–50 μM. LOD competes well with other electrodes reported in the literature. The developed sensor demonstrated good practical applicability for DA in a DA injection with good resultant recovery percentages and RSDs values.

## 1. Introduction

Dopamine (3,4-dihydroxyphenethylamine), an organic chemical of the catecholamine neurotransmitters, is one of the most researched neurotransmitters (NTs) because of its major role in the human body such as in the hormonal, renal, central, and cardiovascular systems, and the human metabolism [1,2,3,4,5,6,7]. DA plays an important role within and outside the brain’s rewards system by reinforcing certain behaviour that results in rewards. In addition, DA controls movements, emotional response functions as a vasodilator, and expands the urine output in the pancreas and kidneys by reducing the formation of insulin. However, elevated dopamine concentration in the brain could cause many neurological disorders such as Tourette’s syndrome, restless leg syndrome (RLS), and several illnesses such as drug dependence, schizophrenia, Parkinson’s disease, depression, degenerative diseases, and attention deficit hyperactivity disorder (ADHD) [8,9,10,11,12,13,14]. As a result, maintaining and controlling the high levels of DA in the human body is important. The health import of DA calls for the need to develop a cost-effective, simple, sensitive, and selective assay with a fast response for DA investigation in pharmaceutical samples.

Different assays such as chromatography [15,16], chemiluminescence [17], fluorescence [18], and electrochemistry [19,20,21,22] is employed for DA detection. The use of electrochemistry has found wide attraction among the various methods owing to high selectivity, sensitivity, simple instrumentation with a fast response, and cost-effectiveness [19,20,21,22]. Iron (III) oxide (Fe_3_O_4_) nanoparticles are one of the most researched forms of iron oxides due to their attractive characteristics, which include low toxicity, biocompatibility, super magnetism, high surface area, and low cost [23]. Fe_3_O_4_ nanoparticles have found applications in catalysis, magnetic resonance imaging contrast, lithium batteries, antibacterial studies, and the removal of heavy metals such as arsenic in water [24]. Fe_3_O_4_ nanoparticles are prepared via a chemical process [25,26,27] or biological (green) methods [23,24]. SPEEK, a non-fluorinated polymer, has found application in electrochemical studies owing to its properties such as excellent stability (chemical, thermal, and mechanical), good thermo-oxidative resistance, and proton conductivity [28]. Polymer metal oxide composites are used as electrode modifiers in electrochemical studies, because of the interactions of intrinsic properties of metal oxide nanoparticles and polymers with increased electrical conductivity and stability [29]. For instance, composites of zinc oxide-sulphonated polyether ether ketone (SPEEK/ZnO), polyaniline–iron (III) oxide (PANI/Fe_3_O_4_) [19,20,21,22], polyaniline-bismuth oxide (PANI-Bi_2_O_3_) [30], polypyrrole-iron (III) oxide (PPy/Fe_3_O_4_) [29], polyaniline-binary metal oxide (NiO/CuO/PANI) [30], and polypyrrole-tunsgten oxide (PPy-WO_3_) [31] were applied as electrode materials for the sensing of dopamine (DA), pramipexole, serotonin, glucose, and gas (hydrogen sulphide) accordingly. Polypyrrole-titanium oxide (PPy-TiO_2_) and polyaniline–zinc oxide (PANI/ZnO) nanocomposites were employed for light-emitting diodes and corrosion protection, respectively [32,33]. Moreover, SPEEK/TiO_2_ was employed in the fabrication of electrodes for fuel cells [28]. In addition, nanostructured carbon black was applied in DA detection [34].

This study, for the first time, reports dopamine electroxidation at the sulphonated polyether ether ketone-Iron (III) oxide modified screen-printed carbon electrode (SPCE-Fe_3_O_4_/SPEEK). The Fe_3_O_4_ nanoparticle (Fe_3_O_4_NPs) was synthesized through a green route (from *L. serica* leaf extract) because the method is environmentally safe, less toxic, and cheap since materials are naturally available. There is no report on nanoparticles synthesized from the *L. serica* leaf. The nanocomposite-modified electrode displayed a well-defined redox voltammogram in the redox probe and good electrocatalytic oxidation of dopamine than the bare SPCE. It also had a good detection limit, selectivity, and was successfully utilized to determine DA in the pharmaceutical sample. Fabrication of the electrode was simple, convenient, and economical.

## 2. Results

### 2.1. Characterization of Fe_3_O_4_ and Fe_3_O_4_/SPEEK

#### 2.1.1. UV–Visible Study

The formation of Fe_3_O_4_NPs is ascribed to ferrous, ferric salts (iron (II) chloride tetrahydrate, iron (III) chloride tetrahydrate, and the leaf extract of *L. serica*. The reduction that occurred on the Fe^2+^ ions is explained by the visible colour change in the reaction mixture which physically confirms the Fe-O nanoparticles by using a UV–Visible spectrophotometer. Figure 1 shows the UV–visible spectra of green mediated Fe_3_O_4_NPs with an absorbed peak of approximately 296 nm, which is close to absorption peaks (296, 259, and 282) reported in previous studies [35]. However, the obtained results show a great biomolecule capping surface of the Fe_3_O_4_NPs without the presence of a Plasmon resonance surface. The energy band gap was calculated to be 4.19 eV according to Equation (1) using the obtained maximum absorption peak (296 nm):(1)$$E_{\mathrm{bg}}=\frac{1240}{\lambda }\;\left(\mathrm{eV}\right)$$

#### 2.1.2. FTIR Study

FTIR spectra of SPEEK, Fe_3_O_4_NPs, and Fe_3_O_4_/SPEEK nanocomposites recorded on the wavenumber from 400–4000 cm^−1^ presented in Figure 2 gives the information on different functional groups of the compounds present. The absorption peak at 3485, 2929, and 2858 cm^−1^ correspond to the –OH stretching of the phenol group and –C-H stretch, which agrees with the literature [36]. The absorption at the 1656, 1596, and 1460 cm^−1^ band corresponds to –C=C stretch which indicates the nitriles group [37]. The 1639 and 1591 cm^−1^ peaks reflect the –C=C stretch aromatic vibrations [38]. The intense peaks at 1090, 1242, and 1215 cm^−1^ are attributed to the –C-O stretch, phenol or alcohol group, and deformation bands in the lignin [37]. The absorption peaks at 469 and 654 cm^−1^ corresponding to Fe-O stretch, confirms the successful synthesis of Fe_3_O_4_NPs, and Fe_3_O_4_/SPEEK nanocomposites, respectively. It is possible that the presence of the phenol –OH group and the amide –N-H group played a role in the reduction of the precursor compound into iron oxide nanoparticles. The polymer SPEEK and iron oxide nanoparticles showed a significant interaction as shown in the composite peaks absorbed (Figure 2).

#### 2.1.3. XRD Study

The crystallographic structure of the samples can be determined using XRD. Figure 3 shows the XRD pattern of Fe_3_O_4_NPs with diffraction peaks and their corresponding planes at 2thetha (θ) values of 26.75° (120), 35.16° (200), 39.22° (123), 52.01° (115), and 55.99° (122), which are similar to a reported study [39]. The diffraction peaks observed confirm the crystalline nature of Fe_3_O_4_ nanoparticles, and the planes of the magnetite Fe_3_O_4_NPs also confirm the rhombohedral hematite phase. The diffraction peak obtained in the XRD patterns indicates that there was no trace of additional planes observed, which indicates the mediated Fe_3_O_4_NPs were obtained in high purity at room temperature [40,41].

#### 2.1.4. SEM Study

The surface morphology of the prepared Fe_3_O_4_NPs from the green synthesis of *L. serica* leaf extract was analyzed by scanning electron microscopy (SEM). Figure 4a,b represent the SEM images of Fe_3_O_4_NPs and Fe_3_O_4_/SPEEK nanocomposites, respectively. Figure 4a depicts the morphology of Fe_3_O_4_NPs that appears to be roughly agglomerated spherical particles in shape, which could be due to the steric effect associated with the magnetic Fe_3_O_4_NPs surface interaction by the active sites [42]. Figure 4b shows the morphology of the Fe_3_O_4_/SPEEK nanocomposites that appeared to have some crystal-like structure which indicates the presence of the SPEEK polymer in the nanocomposite, indicating the occurrence of interaction between Fe_3_O_4_ and SPEEK. Hence, particles appeared to be clustered together, thus, maintaining the agglomerated spherical shape [43].

### 2.2. Electrochemical Characterization

#### 2.2.1. Electrochemical Characterization of Electrodes

The electrochemical efficiency and electron transport properties of electrodes (bare-SPCE, SPCE-Fe_3_O_4_NPs, SPCE-SPEEK, and SPCE-Fe_3_O_4_/SPEEK nanocomposites) were investigated using cyclic voltammetry (CV) at a scan rate of 25 mV/s within −0.2–1.0 V potential window in 0.1 M PBS of pH 7.4 containing 5 mM K_3_[Fe(CN)_6_]. A comparative cyclic voltammogram of the electrodes is presented in Figure 5. The current response was enhanced at the SPCE-SPEEK and SPCE-Fe_3_O_4_/SPEEK electrodes as opposed to the bare and nanoparticle-modified electrodes, which could be due to the presence of SPEEK which has excellent electrocatalytic properties. Table 1 shows the parameters measured at the electrodes.

#### 2.2.2. Scan Rate Study at SPCE-Fe_3_O_4_/SPEEK Electrode

The effect of scan rate variation on peak currents of modified SPCE-Fe_3_O_4_/SPEEK in 5 mM prepared in 0.1 M PBS, pH 7.4 solution was studied using CV in the range from 25–500 mV/s scan rate as shown in Figure 6a. As the scan rate increases, the oxidation peak potentials shifted to the more positive. In consequence, a linear plot of peak currents versus square root of scan rate (v^1/2^) was deduced (Figure 6b). The graph clearly shows an increase of the peak currents with an increase in the square root of the scan rate, indicating a diffusion-controlled electrochemical process, which was also confirmed by the correlation coefficient (R^2^) value of 0.99 [44]. The surface area of the modified nanocomposite electrode (SPCE-Fe_3_O_4_/SPEEK) was found to be 2.799 cm^2^, using a Randle–Sevcik Equation (2) which is higher than the geometry of the bare SPCE (0.125 cm^2^):(2)$$I_{p}=\left(2.69\times 10^{5}\right)n^{\frac{3}{2}}{\mathrm{AD}}^{\frac{1}{2}}{\mathrm{Cv}}^{\frac{1}{2}}\;n$$
where I_p_ represents peak current (A), *n* is the number of electron transfer, A represents surface area (cm^2^), D represents diffusion coefficient (cm^2^/s), C represents concentration (mol/cm^3^), and v represents scan rate (V/s).

From the cyclic voltammetric measurement in Figure 6a, a linear plot of potential peaks (E_pa_/E_pc_) versus the log of scan rate (Figure 7) gave two straight lines with slopes of equal Equations (3) and (4):(3)$$E_{\mathrm{pa}}=\frac{2.303\mathrm{RT}}{\left(\;1-\;\alpha \right)\mathrm{nF}}\;\log\;v$$
(4)$$E_{\mathrm{pc}}=-\frac{2.303\mathrm{RT}}{\alpha \mathrm{nF}}\;\log\;v$$

According to Laviron’s Equations (3) and (4), the number of electron transfer (*n*) and charge transfers coefficient (α) were calculated to be 1 and 0.53, respectively. Additionally, the Tafel value (b) was found to be 0.375 Vdec^−1^ using Equation (5) which is higher than the theoretical value (0.118 Vdec^−1^), suggesting adsorption on the electrode surface by reactants:(5)$$E_{p}=(\frac{b}{2})\;\mathrm{logv}+\mathrm{constant}$$

#### 2.2.3. Electrocatalysis of Dopamine

Figure 8 shows the schematic diagram summarizing the electrode chemical modification of the electrode and the response detection of the electrochemical on the Fe_3_O_4_/SPEEK electrode in dopamine prepared in 0.1 M PBS of pH 7.4.

The behaviour of DA on bare and modified screen-printed electrodes (bare-SPCE, SPCE-Fe_3_O_4_NPs, SPCE-SPEEK, and SPCE-Fe_3_O_4_/SPEEK) was studied using cyclic voltammetry at a 25 mV/s scan rate as shown in Figure 9. Redox peaks were observed in all the electrodes. The oxidation peak current of the SPCE-Fe_3_O_4_/SPEEK electrode was slightly higher than the bare, which could be due to the presence of SPEEK ascribed to its good electrical conductivity, but smaller than the SPCE-SPEEK electrode. However, obtained oxidation potential for DA at the SPCE-Fe_3_O_4_/SPEEK electrode was nearer to 0.25 V expected for DA. The high peak current observed on the SPCE/SPEEK electrode, compared with SPCE-Fe_3_O_4_/SPEEK, could be due to the excellent electronic conductivity property of SPEEK, which enhanced the reactivity of Fe_3_O_4_. The Fe_3_O_4_ nanoparticles-modified electrode showed a lower redox peak current than the bare-SPCE on the DA probe, due to the quick assembling of nanoparticles that conduct to larger particles of Fe_3_O_4_, which may crucially reduce the electrochemical properties of the electrode and, thus, be electro-inactive [45,46]. Parameters determined in cyclic voltammetric detection of DA on bare and modified electrodes are shown in Table 2.

#### 2.2.4. Scan Rate Study on Dopamine

In Figure 10a, the electrochemical impact of varying scan rates in the range from 25–400 mV/s on the anodic peak currents of the nanocomposite-modified SPCE-Fe_3_O_4_/SPEEK toward 0.1 mM DA oxidation was investigated using cyclic voltammetry. An increase in the scan rate resulted in shifts of peak potentials to the more positive, and an increase of peak currents, suggesting a diffusion-controlled process. Figure 10b shows the linear plot of peak currents against the square root of scan rate (v^1/2^) with 0.98 regression values for both anodic and cathodic lines (I_pa_ and I_pc_), confirming a diffusion-controlled process for DA oxidation.

Figure 11 represents the linear plot of peak potentials (E_pa_/E_pc_) against the logarithm of scan rate (v). The Tafel slope value was found to be 0.693 V/dec for DA from the slope value of Figure 11 by applying Equation (5). Obtained values were higher than the expected theoretical value of 0.118 V/dec, suggesting adsorption of the reactant on the electrode surface.

### 2.3. Electro-Analysis of DA

#### Concentration Study of DA

The impact of different concentrations on the DA current response was studied using square wave voltammetry as shown in Figure 12a under the optimal parameters of 0.01 V potential step, 0.001 V amplitude, deposition time of 10 s, and frequency of 25 Hz. The result obtained shows the dependence of reduction peak currents of dopamine on increasing DA concentrations (5 to 50 μM). The poorly defined reduction peak current could be due to the nature of the electrode modifiers. The linear relationship between peak currents and DA concentrations (Figure 12b) yielded a linear regression equation of I_pc_ = 0.005087 [DA] + 4.320144, and regression value of 0.98. The detection limit was calculated to be 7.2 μM by applying Equation (6). The LoD competes well with previous works investigated in the literature (Table 3):(6)$$\mathrm{LoD}=\frac{3.3\times \;\mathrm{SD}}{\mathrm{Slope}\;}$$

SD stands for the standard deviation of the peak current, over the slope of the calibrated plot.

Figure 13a,b show the SW voltammogram of DA and UA accordingly with peak potential observed at 0.23 and 0.36 V for the respective analyte. Figure 13c represents the simultaneous detection of DA and UA of the same concentration with potentials noticed at 0.16 V (DA) and 0.31 V (UA). The shifts in the peak potentials and the peak separation (0.15 V) between DA and UA indicate non-interference of the UA signal with that of DA, successful detection of DA in the presence of UA at the designed electrode, and selectivity of the electrode.

### 2.4. Analytical Application of the Proposed Sensor for Determination of DA in Pharmaceutical Sample

The practical applicability of the designed sensor for DA determination was investigated using a diluted DA hydrochloride injection (dopamine HCl-Fresenius 200 mg/5 mL) sample, spiked with different concentrations of DA standards in accordance with well-established standard addition procedure. The results recorded from the SWV measurements under optimum conditions (0.01 potential step, 0.001 amplitude, deposition time 10 s, and frequency of 25 Hz) are summarized in Table 4. Satisfactory recoveries in the range from 99.9% to 100% were obtained with good relative standard deviations (RSDs), illustrating the promising application of the SPCE-Fe_3_O_4_/SPEEK electrode for the determination of DA in real samples.

### 2.5. Repeatability and Stability Study for Fe_3_O_4_/SPEEK

The repeatability study of the SPCE-Fe_3_O_4_/SPEEK modified electrode was conducted using cyclic voltammetry at a 25 mVs^−1^ scan rate, for 10 repetitive scans in 0.1 mM DA (Figure 14). Relative standard deviations of 0.75 and 2.45% were obtained for oxidation peak potential and peak current, accordingly, suggesting acceptable repeatability, stability, and reproducibility of the electrochemical sensor. Peak current was monitored for 28 days at an interval of 5 days, and the electrode was stored in the refrigerator when not in use. A 45% increase of the initial peak current was observed which could be attributed to increased assimilation of the nanocomposite (SPCE-Fe_3_O_4_/SPEEK) onto the electrode surface over time.

## 3. Materials and Methods

The *L. serica* plant was collected from Kwa-Zulu Natal province. Iron (II) chloride tetrahydrate (FeSO_4_·4H_2_O) and iron (III) chloride hexahydrate (FeCl_3_·6H_2_O) are products from BDH and LABCHEM, South Africa. Sodium phosphate salts (Na_2_HPO_4_ and NaH_2_PO_4_) products of LABCHEM and GlassWorld, South Africa, were used in the preparation of 0.1 M phosphate buffer solution (PBS) of pH 7.4. Potassium ferricyanide (III) (K_3_[Fe(CN)_6_]) and dopamine hydrochloride were purchased from Sigma–Aldrich (St. Louis, MO, USA). Dimethyl formamide (DMF), sodium hydroxide (NaOH), and distilled water was produced by Emplura^®^ Merck (The Chemical Center from Maharashtra, India). All chemicals used were of analytical grade.

### 3.1. Preparation of Plant Leaf Extract

Approximately 10 g of ground fine powdered leaves of the *L. serica* plant was weighed and transferred into a conical flask followed by the addition of 200 mL of distilled water and heated for several minutes at 60 °C until a change in colour (dark green-brown solution) was observed. Leaf extract was filtered using Whatman No. 1 filter paper and a Buchner flask [51].

### 3.2. Synthesis of Iron Oxide Nanoparticles

The green meditation of Fe_3_O_4_NPs derived from *L. serica* leaf extract following the prescribed method with a few minor changes [52,53]. 2:1 M volume ratio of iron (II) chloride tetrahydrate and iron (III) chloride tetrahydrate solution was added to the *L. serica* extract with a resultant black-coloured precipitate, indicating the formation of precipitates (iron oxide nanoparticle). The pH of the mixture was adjusted to 11 by the addition (drop-wise) of 1.0 M of NaOH solution under continuous stirring. The solution was thereafter stirred for 1 h to complete the reaction homogeneity, filtered using vacuum filtered precipitates (Fe_3_O_4_NPs) washed several times with distilled water, and air-dried in the fume wood overnight. The dried sample was stored in an airtight container for further characterization.

### 3.3. Preparation of SPEEK Polymer

Details on SPEEK preparation (sulphonation of polyether ether ketone) have been reported in our previous work [54].

### 3.4. Synthesis of Iron Oxide/SPEEK Nanocomposites

20 mg of SPEEK and 40 mg of iron oxide nanoparticles were dissolved in N, N-dimethylformamide solution, sonicated for 48 h at room temperature, and stored for further characterization.

### 3.5. Characterization of Nanomaterials Synthesized

The successful synthesis of the Fe_3_O_4_NPs and nanocomposite (SPEEK/Fe_3_O_4_) were confirmed through characterizing techniques by using Carry 300, UV–Vis Spectrophotometer, Agilent Technologies, Waldbronn, Germany, spectroquant Prove300, Merck KGaA, (Darmstadt, Germany), and UV–Vis Uviline 9400 (Sl Analytics, Hattenbergstr.10, D-55122 Mainz, Germany) in the investigation of the Nanomaterials optical properties of the nanomaterial fabricated. FTIR (Opus Alpha-P, Brucker Corporation, Billerica, MA, USA). Quanta FEG 250 ESEM, (ThermoFisher Scientific, Waltham, MA, USA) operating on an acceleration voltage of 15.0 kV was employed to describe the surface structure of the nanomaterials prepared. X-ray diffraction spectroscopy (XRD) from Bruker company, Karlsruhe, Germany, and scanning electron microscopy (SEM) from JEOL company, Dearborn, Peabody, MA, USA).

### 3.6. Electrode Modification and Electrochemical Studies

Separate suspensions of Fe_3_O_4_NPs (20 mg), SPEEK (20 mg), and Fe_3_O_4_/SPEEK (20 mg each) were dispersed in DMF and ultra-sonicated for 48 h to form a paste prior to electrode modification. Formed pastes were dropped on SPCE and air-dried to give SPCE-SPEEK, SPCE-Fe_3_O_4_NP, and SPCE-Fe_3_O_4_/SPEEK. Electrochemical studies were conducted on the screen-printed carbon electrode (DropSens 110) of 4 mm in diameter, which consists of working, reference (A/AgCl), and a counter electrode adapted into a Dropview 200 potentiostat powered by Dropview 200 software obtained from Metrohm. Electrochemical techniques employed were cyclic voltammetry (CV) and square wave voltammetry (SWV).

### 3.7. Preparation of Real Sample for Analysis

A dopamine hydrochloride injection (the mL taken) was diluted with distilled water in a 100 mL flask and 2 mL each of the diluted solution was transferred into six 50 mL volumetric flasks. Five of the flasks were spiked with different concentrations of DA stock solution while the sixth flask was held as a control. The flasks were made to the mark using 0.1 M PBS of pH 7.4, and analyzed using SWV in triplicate.

## 4. Conclusions

In this study, the synthesis of Fe_3_O_4_NPs from the *L. serica* plant and the fabrication of nanocomposite-modified SPCE (SPCE/SPEEK/Fe_3_O_4_) for DA detection is reported. The amplified SPCE-Fe_3_O_4_/SPEEK peak current, in contrast to Fe_3_O_4_NPs, could be attributed to the presence of SPEEK. The plot of peak currents versus the square root of scan rate gave a 0.98 regression value, suggesting the occurrence of a diffusion-controlled electrochemical process. The calculated detection limit competes well with previous studies investigated. In addition, the proposed sensor was selective to DA in the presence of uric acid (UA) and yielded good recovery with excellent RSDs in real sample sensing of DA. The results suggest the potential application of the designed sensor for DA monitoring in the pharmaceutical sample.

## Figures and Tables

**Figure 1 molecules-26-05357-f001:**
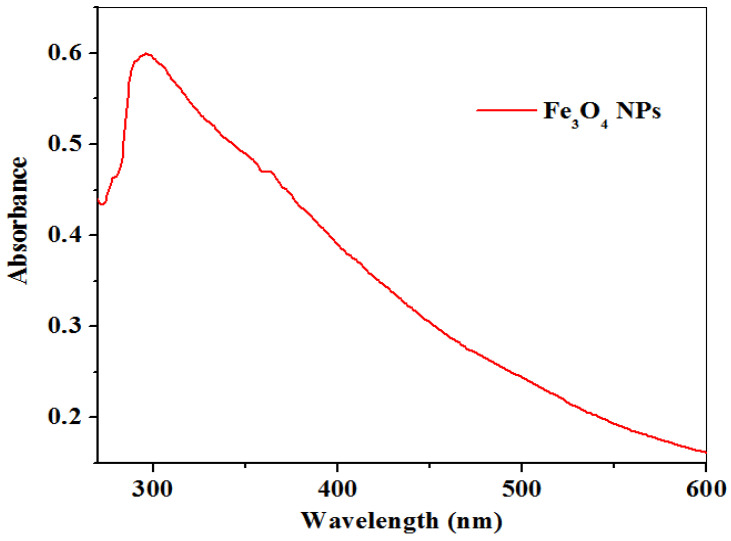
UV–Vis spectra of Fe_3_O_4_NPs using *L. serica* leaf extract.

**Figure 2 molecules-26-05357-f002:**
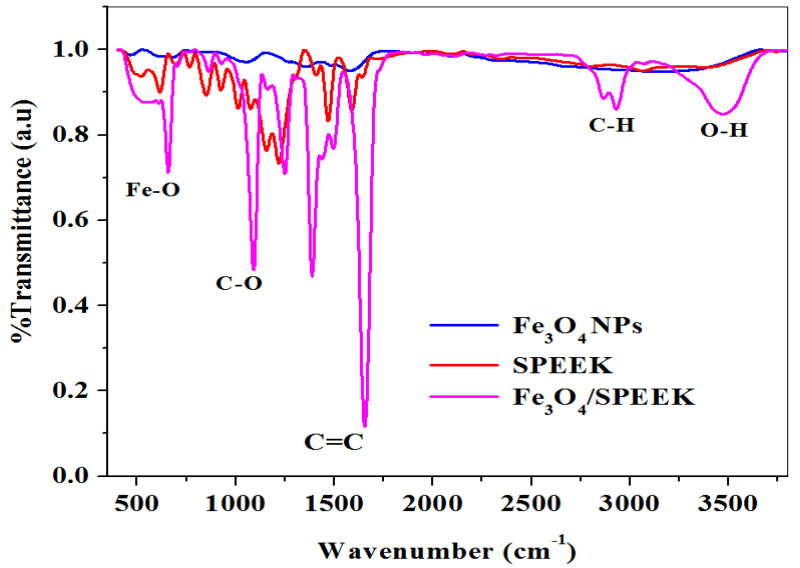
FTIR spectra showing mediated Fe_3_O_4_NPs, polymer SPEEK and Fe_3_O_4_/SPEEK nanocomposites respectively.

**Figure 3 molecules-26-05357-f003:**
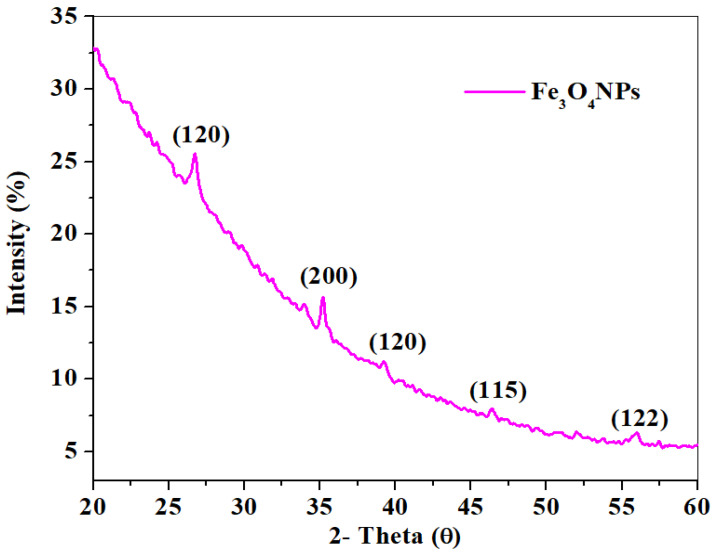
XRD pattern of mediated Fe_3_O_4_NPs using *L. serica* plant species leaf extract.

**Figure 4 molecules-26-05357-f004:**
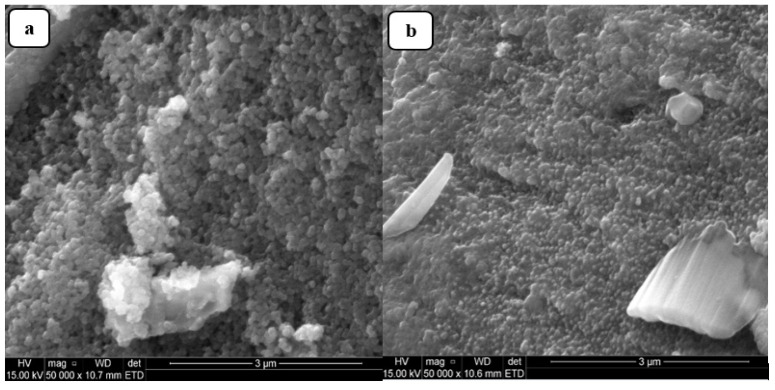
SEM micrographs of (**a**) Fe_3_O_4_NPs and (**b**) nanocomposite (Fe_3_O_4_/SPEEK).

**Figure 5 molecules-26-05357-f005:**
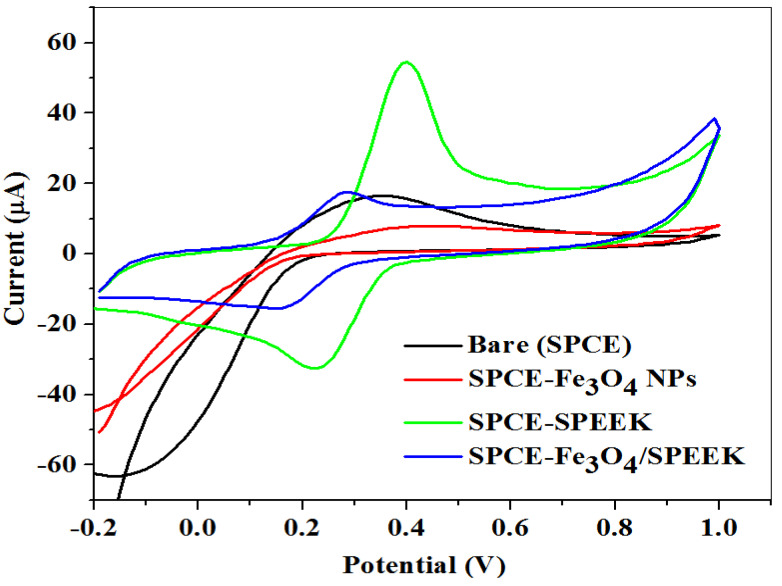
Cyclic voltammogram of the bare and modified electrodes in 5 mM K_3_[Fe(CN)_6_] prepared in 0.1 M PBS, pH 7.4.

**Figure 6 molecules-26-05357-f006:**
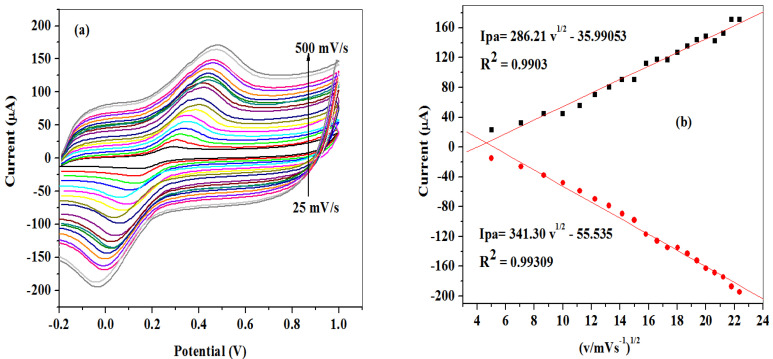
(**a**) Scan rate cyclic voltammograms of SPCE-Fe_3_O_4_/SPEEK. (**b**) Regression plot of peak currents versus square of scan rate in 5 mM K_3_[Fe(CN)_6_] prepared in 0.1 M PBS (pH 7.4).

**Figure 7 molecules-26-05357-f007:**
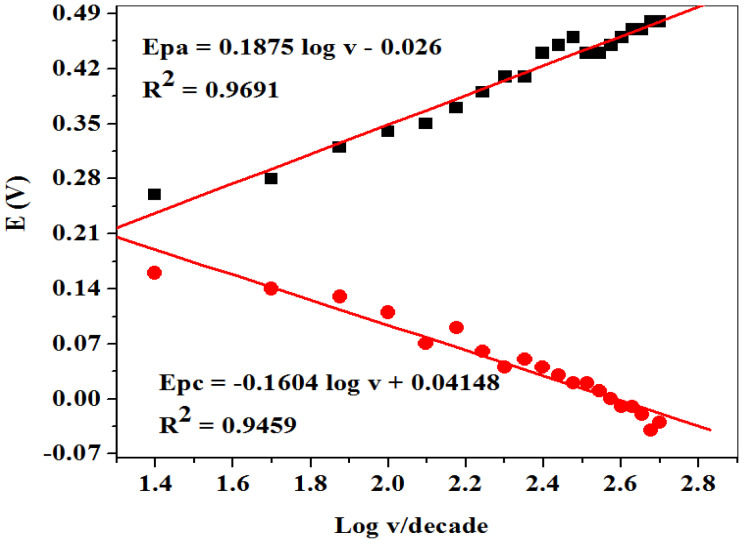
Linear plot of peak potentials (E_pa_/E_pc_) vs. log of scan rate (v/decade) in 0.1 M PBS pH 7.4 containing 5 mM K_3_[Fe(CN)_6_].

**Figure 8 molecules-26-05357-f008:**
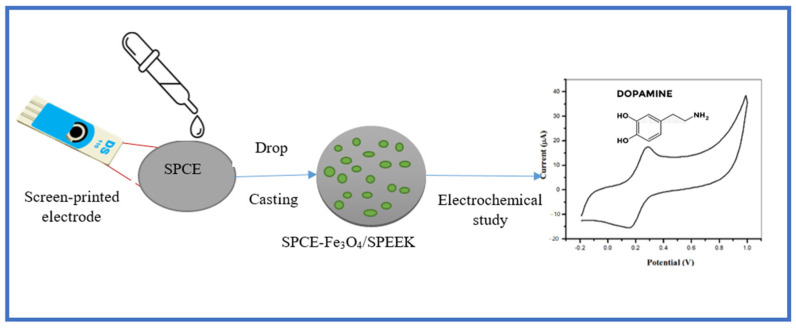
Schematic diagram of electrode modification procedure and the electrochemical response of dopamine at the electrode.

**Figure 9 molecules-26-05357-f009:**
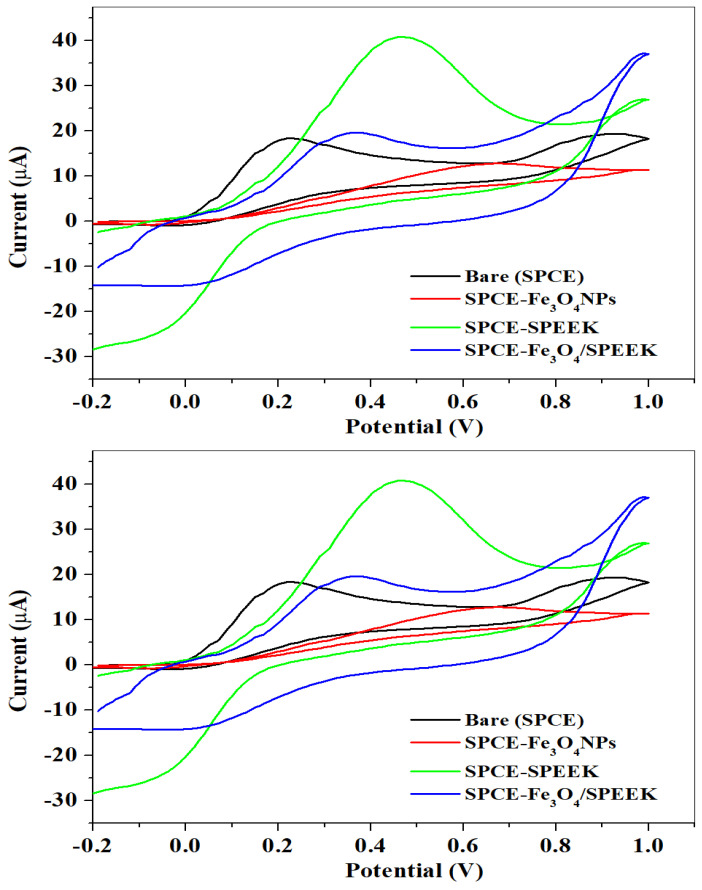
Comparative cyclic voltammogram of bare SPCE and modified electrodes in 0.1 M PBS (pH 7.4) containing 0.1 mM DA.

**Figure 10 molecules-26-05357-f010:**
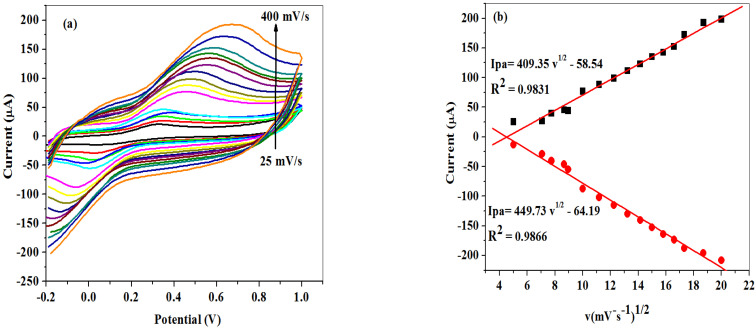
Scan rate cyclic voltammograms at SPCE-Fe_3_O_4_/SPEEK for (**a**) DA and (**b**) linear plots of peak currents (μA) versus square of scan rate in 0.1 M PBS (pH 7.4) containing 0.1 mM DA.

**Figure 11 molecules-26-05357-f011:**
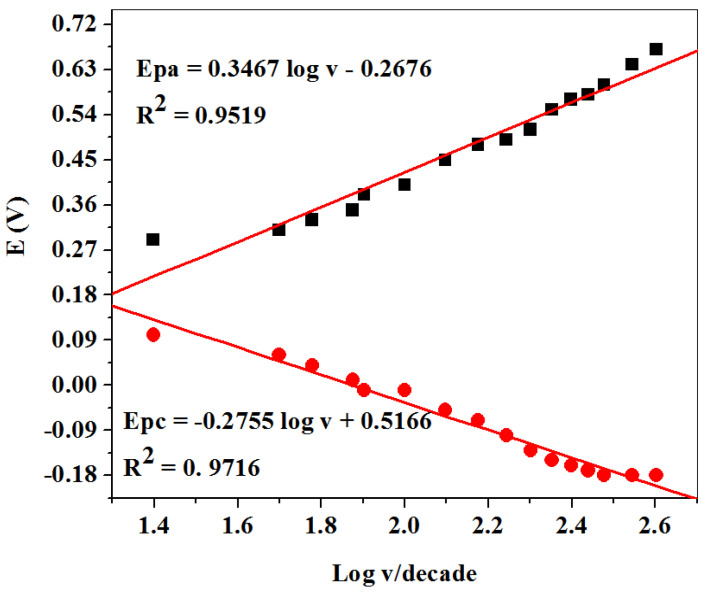
Linear plots of peak potential vs. logarithm of scan rate at SPCE-Fe_3_O_4_/SPEEK for DA prepared in 0.1 M PBS (pH 7.4).

**Figure 12 molecules-26-05357-f012:**
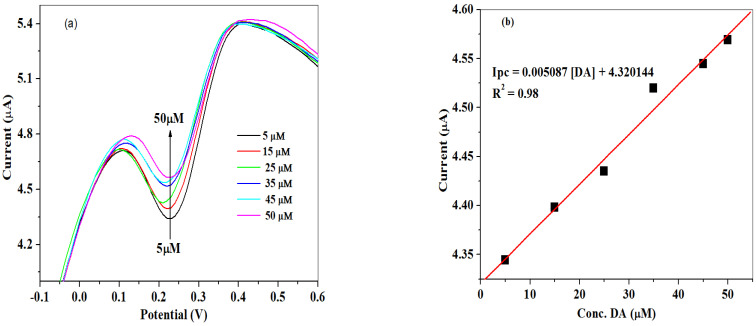
(**a**) Square wave voltammograms recorded at SPCE-Fe_3_O_4_/SPEEK electrode over dopamine concentrations range from 5 to 50 μM; (**b**) linear graph of reduction peak currents versus DA concentrations prepared in 0.1 M of PBS (pH 7.4).

**Figure 13 molecules-26-05357-f013:**
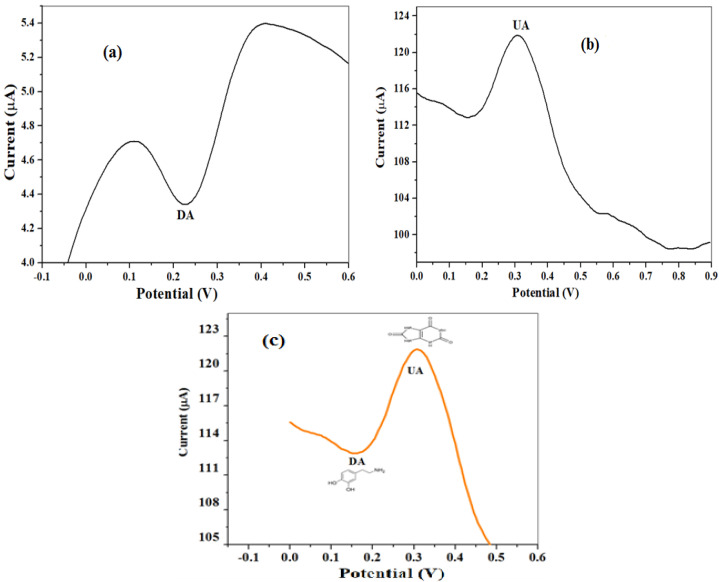
The SWV voltammogram for (**a**) 0.1 mM DA, (**b**) 0.1mM, and (**c**) mixture of DA and UA detected simultaneously at fixed concentration (0.1 mM).

**Figure 14 molecules-26-05357-f014:**
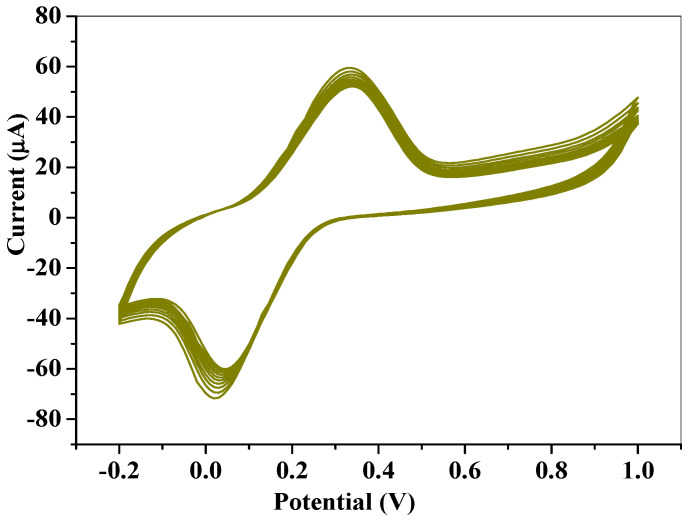
10 repetitive CV scans in 0.1 mM DA at 25 mVs^−1^ scan rate.

**Table 1 molecules-26-05357-t001:** Summary of cyclic voltammetric data recorded at different unmodified and modified electrodes in K_3_[Fe(CN)_6_] solution.

Electrode	I_pa_ (μA)	I_pc_ (μA)	I_pa_/I_pc_	E_pa_ (V)	E_pc_ (V)	E^1/2^ (V)	ΔE_p_ (V)
Bare-SPCE	16.40	−16.97	−0.26	0.36	−0.16	0.10	0.52
SPCE-Fe_3_O_4_NPs	7.99	−40.56	−0.11	0.42	−0.14	0.14	0.56
SPCE-SPEEK	54.48	−32.69	−1.67	0.31	0.22	0.26	0.09
SPCE-Fe_3_O_4_/SPEEK	17.93	−15.18	−1.18	0.29	0.18	0.22	0.11

Where I_pa_ = Anodic peak current, I_pc_ = Cathodic peak current, E_pa_ = Anodic peak potential, E_pc_ = Cathodic peak potential, $E^{1/2}=\frac{E_{\mathrm{pa}}+E_{\mathrm{pc}}}{2},\;\mathrm{and}\;$ ∆E_p_ = Peak potential separation.

**Table 2 molecules-26-05357-t002:** Summary of parameters obtained using cyclic voltammetry on bare and modified electrodes in 0.1 mM DA.

Dopamine	I_pa_ (μA)	I_pc_ (μA)	I_pa_/I_pc_	E_pa_ (V)	E_pc_ (V)	E^1/2^ (V)	ΔE_p_ (V)
Bare-SPCE	19.78	−1.12	−17.66	0.19	0.02	0.11	0.17
SPCE-Fe_3_O_4_NPs	12.56	0.51	24.63	0.66	0.09	0.37	0.57
SPCE-SPEEK	42.00	−25.33	−1.66	0.48	−0.04	0.22	0.52
SPCE-Fe_3_O_4_/SPEEK	20.94	−14.13	−1.48	0.34	0.06	0.20	0.28

**Table 3 molecules-26-05357-t003:** Comparison of the designed sensor with previously studied sensors for DA determination.

Modified Electrode	Methods	Linear Range (μM)	Analyte	LoD (μM)	R^2^	Ref.
Ppy/Ferro-cyanide/carbon paste electrode	LSV	100–1200	DA	38.6	0.9984	[47]
	DPV	200–950	DA	15	0.9998	
Au/Ppy/Ag/GCE	Amperometry	100–5000	DA	50		[48]
*p*-Sulphonatocalix [6]arene/polypyrrole		75–1000	DA	20		[49]
PoPD/E-RGO/GCE		10–400	DA	7.5		[50]
SPCE-Fe_3_O_4_/SPEEK	SWV	5–50	DA	7.1	0.9831	This work

Abbreviations: Ppy = Polypyrrole; Au = Gold; Ag = Silver; E-RGO = Electrochemically-reduced graphene oxide; GCE = Glass carbon electrode, PoPD = poly(o-phenylenediamine).

**Table 4 molecules-26-05357-t004:** Results of the recovery tests obtained from the DA determination using dopamine HCl-Fresenius 200 mg/5 mL injection.

Sample	Added (μM)	Detected (μM)	Recovery (%)	RSD
Dopamine HCl-Fresenius 200 mg/5 mL injection	40	39.97	99.9	3.90
	80	80.01	100	0.05
	120	119.4	99.9	0.12

## Data Availability

The data presented in this study are available on request from the corresponding author.

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
