# Peer review of "Electrochemical Detection of Dopamine at Fe3O4/SPEEK Modified Electrode"

_molecules, 2021, doi:10.3390/molecules26175357_

Round 1

Reviewer 1 Report

The authors did make modifications based on my last comments, although some part was not well answered (the selectivity test, for example), they did explain that was due to the lack of time and covid restrictions.  

There’re still some issues left in this revision.

  1. In Table 1, why is the Epa more negative than Epc for SPCE-Fe3O4/SPEEK ?
  2. Are the CVs in Fig. 9 another set of data, or the original data but smoothened? If they’re the smoothened data, the author should clarify it in the manuscript. And technically, they should collect better quality data instead of smoothen the original data, which I understand is due to time restriction and covid issues.
  3. The authors mentioned that SPCE-Fe3O4-SPEEK electrodes have better results in liability and concentration studies than SPCE- SPEEK electrodes, it’s probably better to show some of the data (for example the R2 in Table 3), because this is the advantage of SPCE-Fe3O4-SPEEK electrodes.

Author Response

The Editor

Molecules

REVISION OF MANUSCRIPT SUBMITTED FOR PUBLICATION

Manuscript Title: Electrochemical detection of dopamine at Fe3O4/SPEEK modified electrode

Manuscript ID: molecules-1349319

Thank you for reviewers reports on our manuscripts. I hereby submit the revised version of the manuscript for further consideration.

We appreciate the editor and reviewers for painstakingly going through the manuscript and making such comments and suggestions that have helped in improving the quality of the manuscript. All the comments and suggestions from reviewers have been carefully considered and treated accordingly.

Please find below, authors’ responses to the reviewers’ comments and suggestions. Comments and suggestions from the reviewers are written first with the heading “Reviewer’s remark” followed by authors’ responses with the heading “Authors’ response”. Kindly also note that the revised areas of the manuscript are highlighted yellow in the revised manuscript.

Reviewer 1

Reviewer’s remark

  1. In Table 1, why is the Epa more negative than Epc for SPCE-Fe3O4/SPEEK?

Authors’ response

Thanks for the observation, the Epa and Epc values were mistakenly swapped around. Kindly refer to the yellow highlights on Table 1 page 6.

Reviewer’s remark

  1. Are the CVs in Fig. 9 another set of data, or the original data but smoothened? If they’re the smoothened data, the author should clarify it in the manuscript. And technically, they should collect better quality data instead of smoothen the original data, which I understand is due to time restriction and covid issues.

Authors’ response

Thanks for the observation. Kindly refer to Figure 9 has been improved in page 9 and 10. The original data have been smoothened to reduce some bumps on all the graphs.

Reviewer’s remark

  1. The authors mentioned that SPCE-Fe3O4-SPEEK electrodes have better results in liability and concentration studies than SPCE- SPEEK electrodes, it’s probably better to show some of the data (for example the R2in Table 3), because this is the advantage of SPCE-Fe3O4-SPEEK electrodes

Authors’ response

         Thank so much for your comments. The values of R2 have been included on Table 3, please refer to the yellow highlights in Table 3 page 12. But for references 48, 49 and 50, no R2 values were reported in these literatures.  

Reviewer 2 Report

Ms. Rita Lin

Assistant Editor

Correspondence reference: molecules-1349319

“Electrochemical detection of dopamine at Fe3O4/SPEEK modified electrode”

Dear Editor,

The manuscript Electrochemical detection of dopamine at Fe3O4/SPEEK modified electrodewas meticulously corrected and improved by the authors. Now, the manuscript can be accepted for publication is its current form.

Author Response

Thanks for the comment.

Regards